# The development and structure of the mesentery

Kevin G. Byrnes[1,2], Dara Walsh [1,2], Leon G. Walsh[1,2], Domhnall M. Coffey[1,2], Muhammad F. Ullah[1,2], Rosa Mirapeix[3], Jill Hikspoors [4], Wouter Lamers [4], Yi Wu[5], Xiao-Qin Zhang[5], Shao-Xiang Zhang[5], Pieter Brama[6], Colum P. Dunne [2], Ian S. O'Brien[7], Colin B. Peirce[1], Martin J. Shelly[8], Tim G. Scanlon[8], Mary E. Luther[1], Hugh D. Brady[1], Peter Dockery[7], Kieran W. McDermott[2] & J. Calvin Coffey [1,2✉]

The position of abdominal organs, and mechanisms by which these are centrally connected, are currently described in peritoneal terms. As part of the peritoneal model of abdominal anatomy, there are multiple mesenteries. Recent findings point to an alternative model in which digestive organs are connected to a single mesentery. Given that direct evidence of this is currently lacking, we investigated the development and shape of the entire mesentery. Here we confirm that, within the abdomen, there is one mesentery in which all abdominal digestive organs develop and remain connected to. We show that all abdominopelvic organs are organised into two, discrete anatomical domains, the mesenteric and non-mesenteric domain. A similar organisation occurs across a range of animal species. The findings clarify the anatomical foundation of the abdomen; at the foundation level, the abdomen comprises a visceral (i.e. mesenteric) and somatic (i.e. musculoskeletal) frame. The organisation at that level is a fundamental order that explains the positional anatomy of all abdominopelvic organs, vasculature and peritoneum. Collectively, the findings provide a novel start point from which to systemically characterise the abdomen and its contents.

[1] Department of Surgery, University of Limerick Hospitals Group, Limerick, Ireland. [2] 4iCentre for Interventions in Infection, Inflammation and Immunology, School of Medicine, University of Limerick, Limerick, Ireland. [3] Department of Anatomy and Embryology, Universitat Autònoma de Barcelona, Barcelona, Spain. [4] Department of Anatomy and Embryology, Maastricht University, Maastricht, Netherlands. [5] Digital Medicine Department, Biomedical Engineering College, Third Military Medical University, Chongqing, China. [6] School of Veterinary Medicine, Veterinary Science Centre, Dublin, Ireland. [7] Department of Anatomy, National University of Ireland Galway, Galway, Ireland. [8] Department of Radiology, University of Limerick Hospitals Group, Limerick, Ireland. ✉email: calvin.coffey@ul.ie

Until recently, the mesenteries were considered peritoneal structures and defined as a double fold of peritoneum connecting some regions of intestine to the posterior abdominal wall. The peritoneum is defined as the serous membrane lining the inner surface of the abdomen. Contemporary findings demonstrated that the mesentery below the duodenum is continuous and the small and large intestine are connected to it[1–3]. Whilst there are suggestions these properties apply above the duodenum, direct evidence is lacking[4]. If they held, it could alter our understanding of the organisation of abdominal contents in general. At present, the position of organs is described in peritoneal terms[5]. In that description, order is maintained by a network of peritoneal derivatives[6]. The proposed network includes mesenteries, omenta, ligaments, folds and membranes[7]. Conventional descriptions of abdominal development maintain that as it develops, the mesentery fragments into mesenteries[8–10]. Some organs thus lose adjoining mesentery to become retroperitoneal. Although the peritoneal model is the standard model of abdominal organisation, it is unclear how this model develops. The model does not explain current surgical techniques or radiological depictions of the abdomen[11–14]. For example, the surgical techniques of total mesorectal excision and complete mesocolic excision are mesenteric-based (not peritoneal-based), and are associated with improved outcomes for patients undergoing surgery for rectal or colon cancer respectively[12,13]. In contrast with the tenets of the peritoneal model, recent findings of mesenteric continuity and connectivity with other organs, reconcile observations across several fields. The findings have been assimilated into reference texts and prompt investigation of the growth and form of the entire abdominal mesentery[1,15,16].

In this study, we determined the program of morphological changes the mesentery undergoes during development to the adult form. We determined the morphology of the adult form. The findings demonstrate the mesentery is the organ in and on which all abdominal digestive organs develop and remain directly connected to. The findings illuminate the composition of the anatomical foundation of the abdomen, and the order at that level. In turn, they explain the positional anatomy of all abdominal digestive organs (including that of the vasculature associated with these) and the organisation of the peritoneum.

## Results

**Mesenteric development.** We reconstructed the developing mesentery from digitized datasets[17]. These were examined to determine the morphology of the mesentery. The accuracy of reconstructions was confirmed using several approaches (Supplementary Note 1, Supplementary Fig. 1). We found the mesentery was continuous and composite at all stages examined (Fig. 1, Supplementary Atlas Sections 1–5, Supplementary Figs. 1, 2). Throughout development, it comprised a highly cellular stroma ("mesodermal mesentery") and surface mesothelium. All abdominal digestive organs developed in or on the mesentery.

To identify key changes in mesenteric morphology, we compared shape at successive time points. A mid-region fold emerged early subdividing the mesentery into upper (pre-fold), mid and lower (post-fold) regions (Fig. 1c–o, Supplementary Note 2 (sections 1–5)). This format of regionalisation was apparent at all subsequent stages (and in the adult (see below)). The mid-region fold provided a continuity between upper and mid-regions (on the right of the superior mesenteric artery (SMA)) and between mid and lower regions (on the left of the SMA). During development, the sides of the fold switched position relative to the SMA (Fig. 1g–o, Supplementary Note 2 (section 4)). Following the switch, the original right side of the fold commenced centrally on the right, but continued peripherally on the left of the SMA. The original left side

commenced peripherally on the right of the SMA but returned centrally to the left. This organisation was also apparent in the adult setting (see below). Detailed descriptions of the morphology of the developing upper, mid and lower regions of the mesentery are included in Supplementary Note 2 (sections 1–5).

Clarification of the shape of the mesentery meant we could characterise the anatomical and histological relationships between it and posterior abdominal wall (Supplementary Note 2 (sections 6, 7), Supplementary Figs. 1, 2). Given digestive organs developed in mesentery, these relationships are relevant to digestive system function. At Carnegie stage (CS) 13, we found each histological element of the mesentery was continuous with a corresponding element in the posterior abdominal wall (Fig. 2). At CS 21, a demarcation was apparent between mesodermal mesentery and posterior abdominal wall in the midline (i.e., these had disconnected but remained apposed) (Fig. 2). Continuity remained at the surface mesothelial level. At later stages, demarcation between mesentery and abdominal wall had extended from the midline laterally, and the mesothelial continuity spanning the surface of both was displaced in tandem[18–20]. The findings indicate that the mesentery does not hinge laterally from a midline junction with the abdominal wall. Instead, it gradually adheres from medial to lateral displacing overlying mesothelium (Supplementary Note 2 (sections 6,7)). Direct in vivo evidence of displacement is apparent after abdominal surgery. Following surgery, mesentery and conjugate organs adhere to the inner surface of the closure site. This occurs from medial to lateral, displacing overlying peritoneum[21,22]. Displacement also explains why digestive organs have peritonealised and non-peritonealised surfaces, as well as the distribution of these.

Histological connection between mesentery and developing organs means that for any developmental stage, the position of a digestive organ (and associated vasculature) can be described in mesenteric terms. Based on this, we clarified the relationship between mesentery and digestive organs at *each* stage (Fig. 3a, Supplementary Note 2 (sections 8–10), Supplementary Atlas (Section 5)). In line with previous findings, we found that digestive organs appeared to differentiate or "emerge" from mesentery[23]. Direct connection between mesentery and other organs enables accurate mapping of cellular and molecular events during mesenteric-based organogenesis[24,25].

The morphological changes identified above explain how the full term foetal mesentery takes shape (Figs. 1–3, Supplementary Note 2 (section 11)). They also explain how abdominal digestive organs are distributed. We found that the shape of the full term foetal mesentery corresponded with those of the in vivo and ex vivo adult mesentery ("Methods", Supplementary Fig. 1). This correlation means that events that shape the foetal mesentery, determine that of the adult (Supplementary Note 2 (section 11)). Using digital reconstructions we generated a video depiction of key events in mesenteric development (Supplementary Atlas (Section 3))[17]. Primary and secondary folding generated a complex picture which could be simplified by digitally removing small intestinal folding (Supplementary Atlas (Section 4)).

**Mesenteric anatomy.** We hypothesised that mesenteric continuity and direct organ connectivity are also apparent in the adult setting. To test this, we excised the abdominal mesentery intact from adult human cadavers (Methods, Supplementary Atlas (Sections 6.1–6.15)). The ex vivo mesentery was continuous from the oesophagogastric junction to the anorectal junction (Fig. 3b–k, Supplementary Atlas (Sections 6.15–6.22)). In addition, all abdominal digestive organs (and spleen) were directly connected to it. They were not connected to the abdominal wall. Of note, mesenteric continuity is implicit in many Renaissance depictions of human anatomy[26–28].

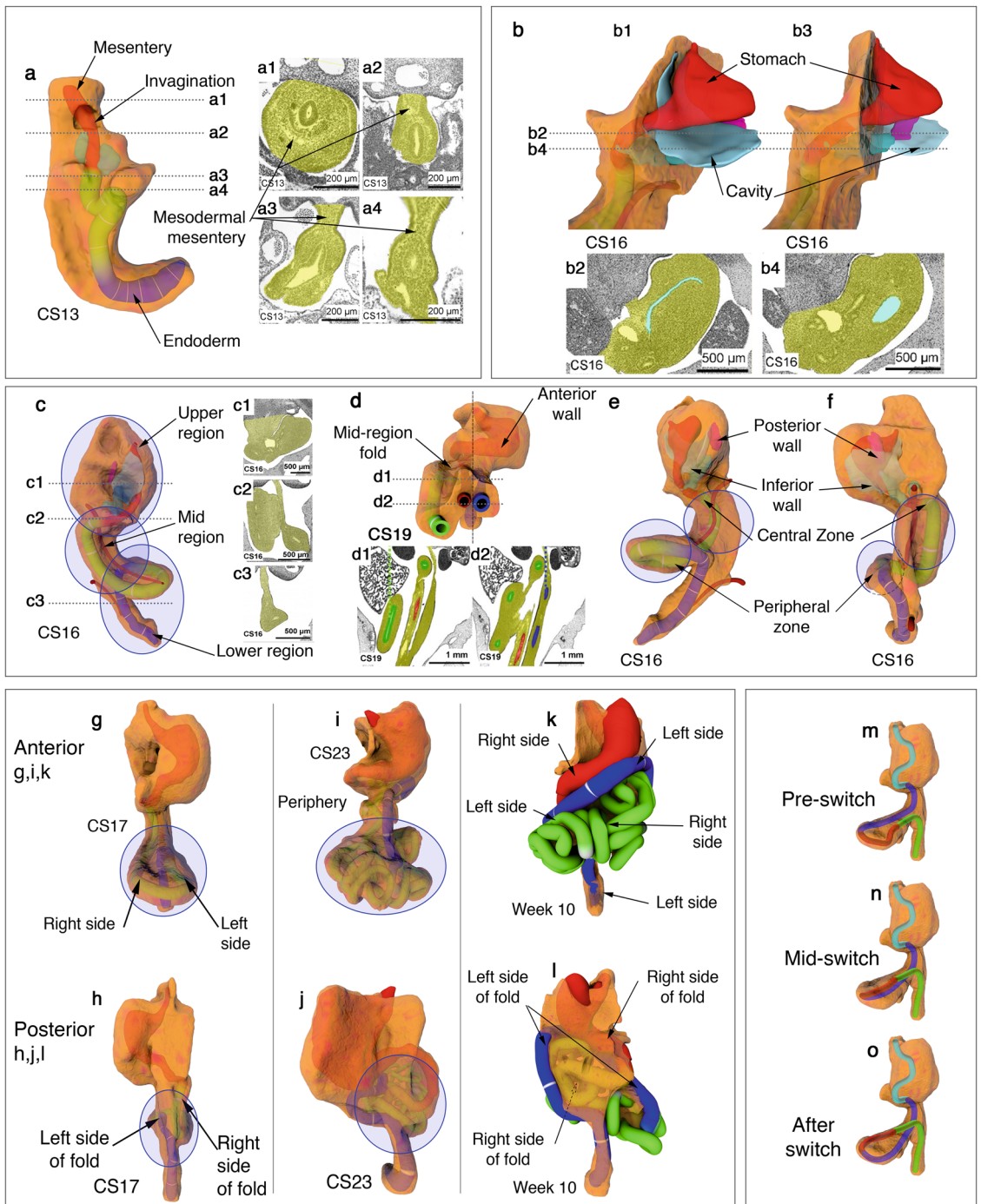

**Fig. 1 Developing mesentery. a** Photograph of a digitally reconstructed mesentery (yellow) at Carnegie stage (CS) 13. **a1–4** Photomicrographs at levels indicated. **b1–3** Photographs of the reconstructed mesentery at CS 16. **b2, 4** Photomicrographs of sections at levels indicated. **c1–4** Photographs and photomicrographs of mesenteric regions at CS16. **d1–2** Photographs and photomicrographs of the anterior aspect of the mid-region fold. Photographs demonstrating anterior (**e**) and posterior (**f**) aspects of the mid-region at CS 16. **g–l** Photographs showing anterior and posterior aspects of the developing mid-region. **m–o** Schematic depiction of the mid-region switch.

We compared the shape of the ex vivo and in vivo mesentery ("Methods"). Mesenteric shape was similar in both settings and matched that of the full term foetus (Supplementary Fig. 1). The shape of the ex vivo mesentery corresponds with observations in in vivo surgical, radiological and pathological settings[29–31]. Collectively, the findings mean that observations related to the ex vivo mesentery, are applicable to the in vivo adult and full term foetal mesentery. Examination of the ex vivo mesentery showed it was subdivided into upper, mid and lower regions by a mid-

region fold. The regional anatomy of the ex vivo mesentery is demonstrated in Supplementary Atlas (Sections 6.18-6.21), and described in Supplementary Note 3 (sections 1–6, Supplementary Figs. 3–6).

We characterised the regional anatomy of the mid-region fold. As was apparent during development, the fold provided a structural continuity between upper and mid regions (on the right of the superior mesenteric artery (SMA)) and between mid and lower regions (on the left of the SMA) (Fig. 4, Supplementary

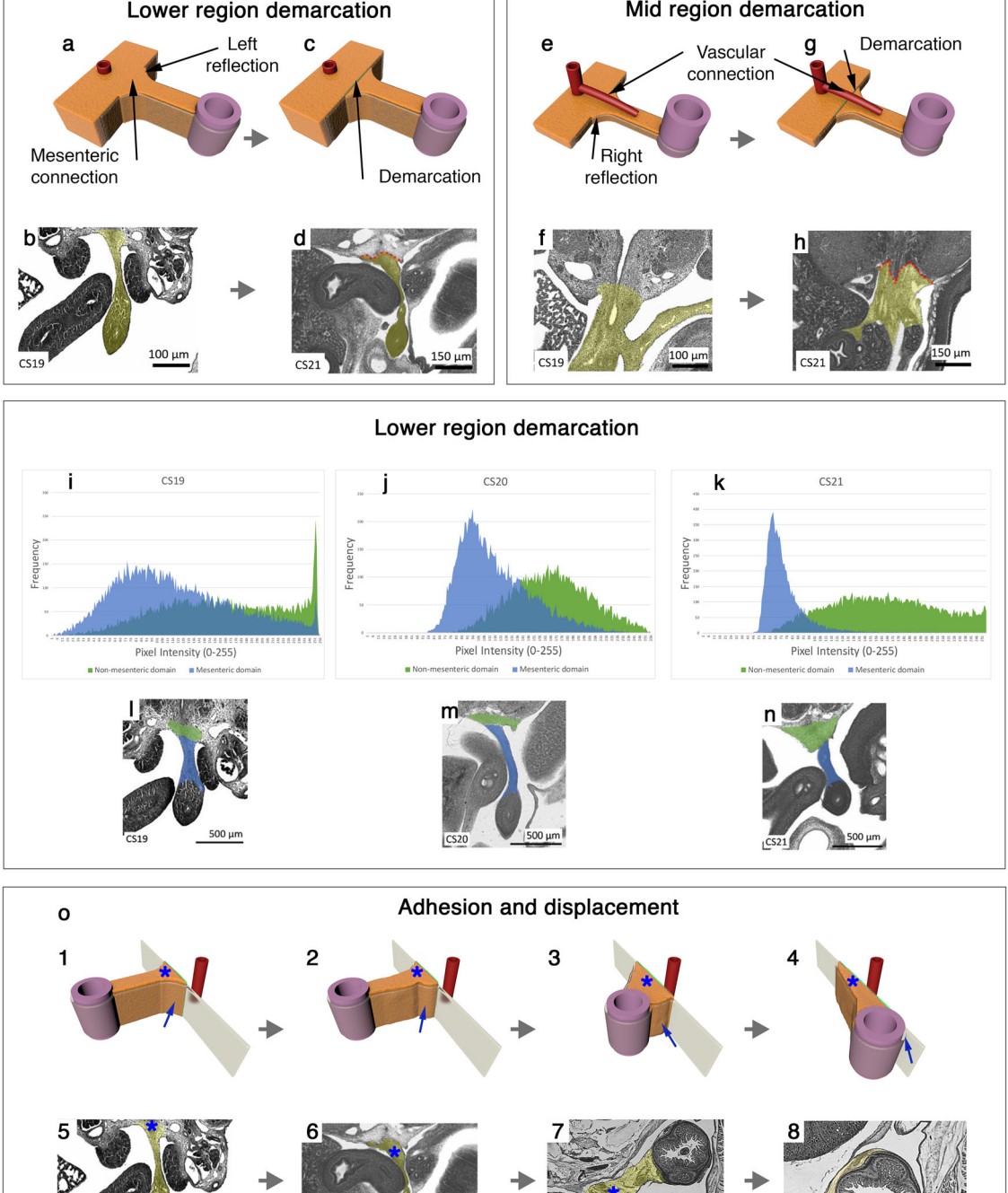

**Fig. 2 The mesentery and posterior abdominal wall. a–d** Photographs and corresponding photomicrographs demonstrating continuity between lower region mesentery and posterior abdominal wall. **e–h** Photographs and corresponding photomicrographs demonstrating continuity between mid-region mesentery and posterior abdominal wall. **i–k** Histograms demonstrating brightness values of developing mesentery and posterior abdominal wall. **l–n** Photomicrographs from which histograms were generated. **o1–4** Panel demonstrating the relationships between the mesentery (*) and posterior abdominal wall during development. The mesothelial junction between both is indicated (**arrow**). **o5–8** Photomicrographs demonstrating histological correlates for o1-4.

Note 3 (section 6)). The sides of the fold switched position relative to the SMA, from central to peripheral zone. A secondary folding was also apparent at the junction between mesoduodenum (i.e., mesentery adjoining the duodenum) and mesojejunum (mesentery adjoining the jejunum). This had several morphological implications (Supplementary Note 3 (section 4)). Given the structural importance of the mid-region switch, we determined if it was also present in vivo ("Methods"). The switch was apparent in reconstructions of the in vivo mesentery and associated vasculature (Supplementary Note 3 (section 5)).

Given all abdominal digestive organs were connected to the mesentery, the relationship between mesentery and abdominal wall is relevant to digestive function (Fig. 5). Prior to its dissection, the mesentery (and conjugate organs) were flattened against the posterior abdominal wall (Supplementary Note 3 (sections 7–10), and Supplementary Atlas (sections 6.1–6.8)). A

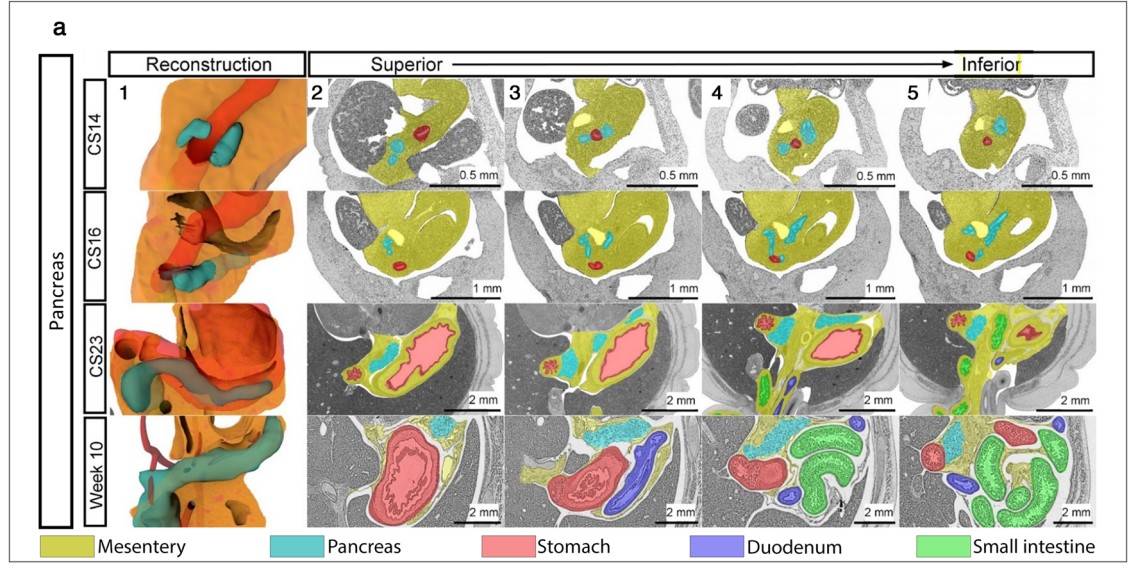

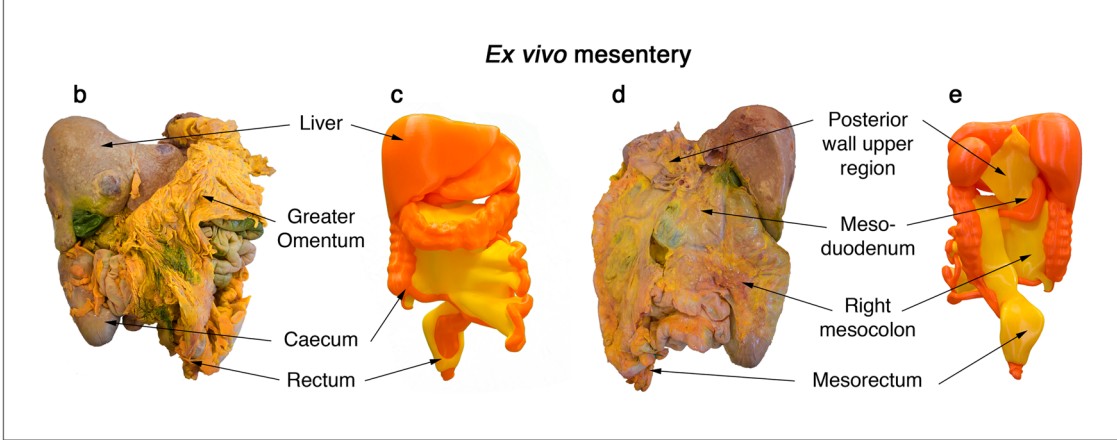

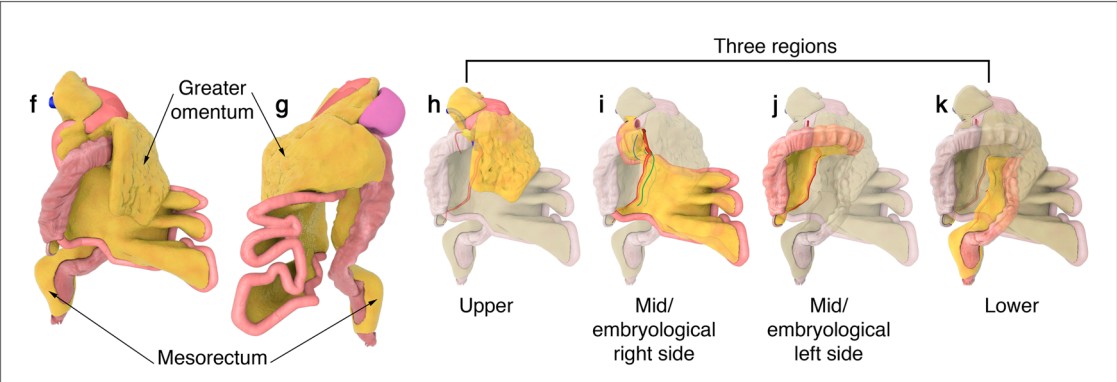

**Fig. 3 Developing and adult mesentery. a** Panel demonstrating the developing mesentery, stomach and pancreas. Each row corresponds to a developmental stage and reads from left to right. **b** Photographs demonstrating the anterior aspect of (**b**) the ex vivo mesentery and of a 3D printed ex vivo mesentery (**c**). Photographs demonstrating the posterior aspect of (**d**) the ex vivo mesentery and of the 3D print used in (**c**, **e**). Illustrations demonstrating the right anterolateral (**f**) and left posterolateral (**g**) aspects of a digital version of the ex vivo mesentery. **h-k** Illustrations of the same digital reconstruction with individual regions highlighted.

fascia occurred between apposed surfaces and a reflection that resembled peritoneum, marked the peripheral limit of apposition (Fig. 5). Collectively, the topographical relationship between mesentery, reflection, fascia and abdominal wall (Fig. 5, Supplementary Atlas (sections 6.16-17)) replicated the organisation of the surface mesothelium, mesentery and posterior abdominal wall during development (Fig. 2). The overlap in organisation, supports the suggestion that the developing mesentery first disconnects from but then progressively adheres to, the posterior abdominal wall. This occurs from medial to lateral and displaces junctional mesothelium.

Given abdominal digestive organs were directly connected to the mesentery, the position of each organ (and associated vasculature) could be described in mesenteric terms. Using this approach, we resolved several longstanding questions in abdominal anatomy (Supplementary Note 3 (sections 11–13), Supplementary Figs. 7, 8).

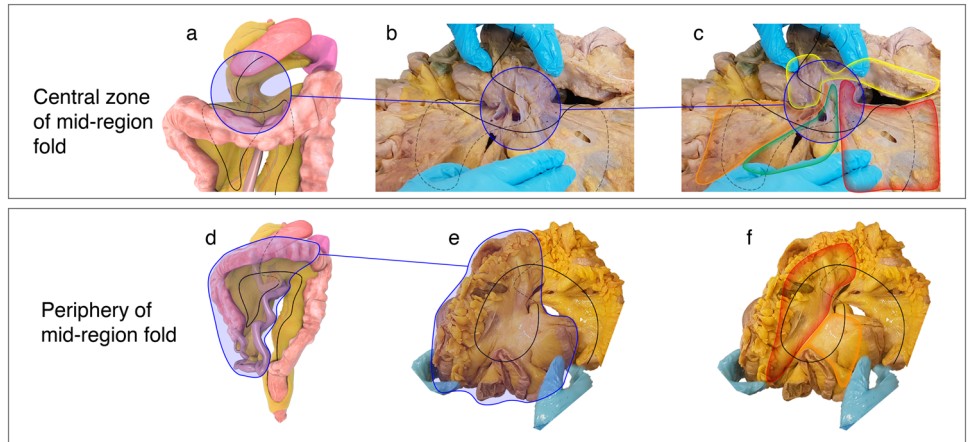

**Fig. 4 Mid region of the mesentery.** Photographs demonstrating the anterior aspect of the central zone (encircled) of the mid-region in (**a**) a digital and (**b**) ex vivo mesentery. In (**c**) a schematic has been superimposed on (**b**) explaining the underlying anatomy. colour code: pancreas (yellow), right mesocolon (orange), hepatic flexure (green), splenic flexure (red). Photographs demonstrating the anterior aspect of the peripheral zone (blue boundary) in (**d**) a digital model and (**e**) the ex vivo mesentery. In (**f**) a schematic has been superimposed on (**e**) explaining the underlying anatomy. colour code: original left side of the peripheral zone (red), original right side of the peripheral zone (orange).

**The mesenteric model of abdominal anatomy**. Collectively, the mesentery and connected organs form a discrete anatomical domain (i.e., the mesenteric domain) (Fig. 5, Supplementary Atlas (section 6.22)). The genitourinary organs are positioned on the musculoskeletal frame of the abdomen. These form a second, discrete domain (the non-mesenteric domain). The findings demonstrate that digestive and genitourinary organs are domain-specific, they develop in and remain on, the same domain. To determine if this modelling held in general, we switched an intact ex vivo mesentery between adult human cadavers. This fully restored organ and tissue-level organisation in the abdomen of each cadaver, albeit with the abdominal digestive system of an alternative cadaver (Fig. 5, Supplementary Atlas (section 6.22)). We next generated a digital replica of the adult human abdomen. When all abdominopelvic organs and associated circuitries (i.e., vascular etc,) were digitally subtracted from both domains, the mesentery and musculoskeletal frame remained. Similar findings emerged when this process was repeated for each stage of development. These findings mean the mesenteric and musculoskeletal frame of each domain, form visceral and somatic components of the anatomical foundation of the abdomen. They mean that the organisation at the foundation is a fundamental order that explains the positional anatomy of all abdominal contents (Supplementary Note 3). This interpretation of abdominal anatomy was arbitrarily termed the mesenteric model.

To determine how this organisation is maintained, we reviewed the anatomical connections between domains. The connections corresponded to the physical links that were disrupted during excision of the mesentery (Supplementary Information; Anatomy 14, Supplementary Atlas (sections 6.16–6.17)). We observed central (i.e. vascular), intermediate (i.e., fascial) and peripheral (reflection) mechanisms. The central and intermediate mechanisms are described in Supplementary Note 3 (section 14). The reflection was located at the periphery of the mesenteric domain where it bridged the peritonealised (i.e., mesothelial) surfaces of both domains. We determined the composition of the reflection and found that it comprised a single layer of peritoneum (Supplementary Fig. 8).

If the mesenteric model held, then it should be possible to explain the positional anatomy of all elements of the abdomen, and at all stages. In the preceding sections, we confirmed this for all organs connected to the mesentery. We determined if the same applied to the peritoneum. Collectively, the findings demonstrated that the peritoneum corresponds to the surface lining of the mesenteric domain (i.e., visceral peritoneum), the non-mesenteric domain (i.e., parietal peritoneum) and the junction between both (i.e., the reflection). Hence, the mesenteric model explains the organisation of the peritoneum.

**Comparative anatomy of the mesentery**. We confirmed that mesenteric continuity and connectivity with abdominal organs were apparent in other animal species (Supplementary Note 3 (section 15), Supplementary Atlas (section 7)). The findings indicate that the anatomical foundation (and order) in the human abdomen, applies to a range of animal species. It is noteworthy that similar observations can be made from Renaissance depictions of abdominal anatomy[32–35]. Indeed, this anatomical foundation is apparent in Coelomata in general, and in fossilised records of these.

**Curve/buckle coupling**. Continuity and connectivity of the intestine and adjoining mesentery provide a mechanical platform on which differential lengthening of both influence gut shape[25,36]. Findings in animal models demonstrate that, in this system, mechanical interactions produce curve and buckle formation in the intestine and mesentery respectively. Understanding the shape of the developing and adult mesentery meant we could now examine for curve/buckle coupling in the human setting. Curve/buckle coupling was apparent at all stages of development, at all levels in the developing abdomen, and in all species examined (Supplementary Fig. 9). Further exploration of curve/buckle coupling resolved multiple properties of intestinal anatomy that hitherto were not fully explained (Supplementary Note 3 (sections 16,17)). To further test the relevance of curve/buckle coupling we developed and tested a conal model of the mesentery ("Methods", Supplementary Fig. 9). The conal model predicts the periphery of the mesentery follows an elliptical path, but deviates from that at junctions with successive cones. We tested this for the mid-region of the ex vivo mesentery. The predictions held, supporting the applicability of the conal model and the importance of simple curve/buckle coupling in determining the shape of the mesentery (Supplementary Fig. 9).

## Discussion

The findings clarify the nature of the mesentery. It is the organ in which all abdominal digestive organs develop and remain

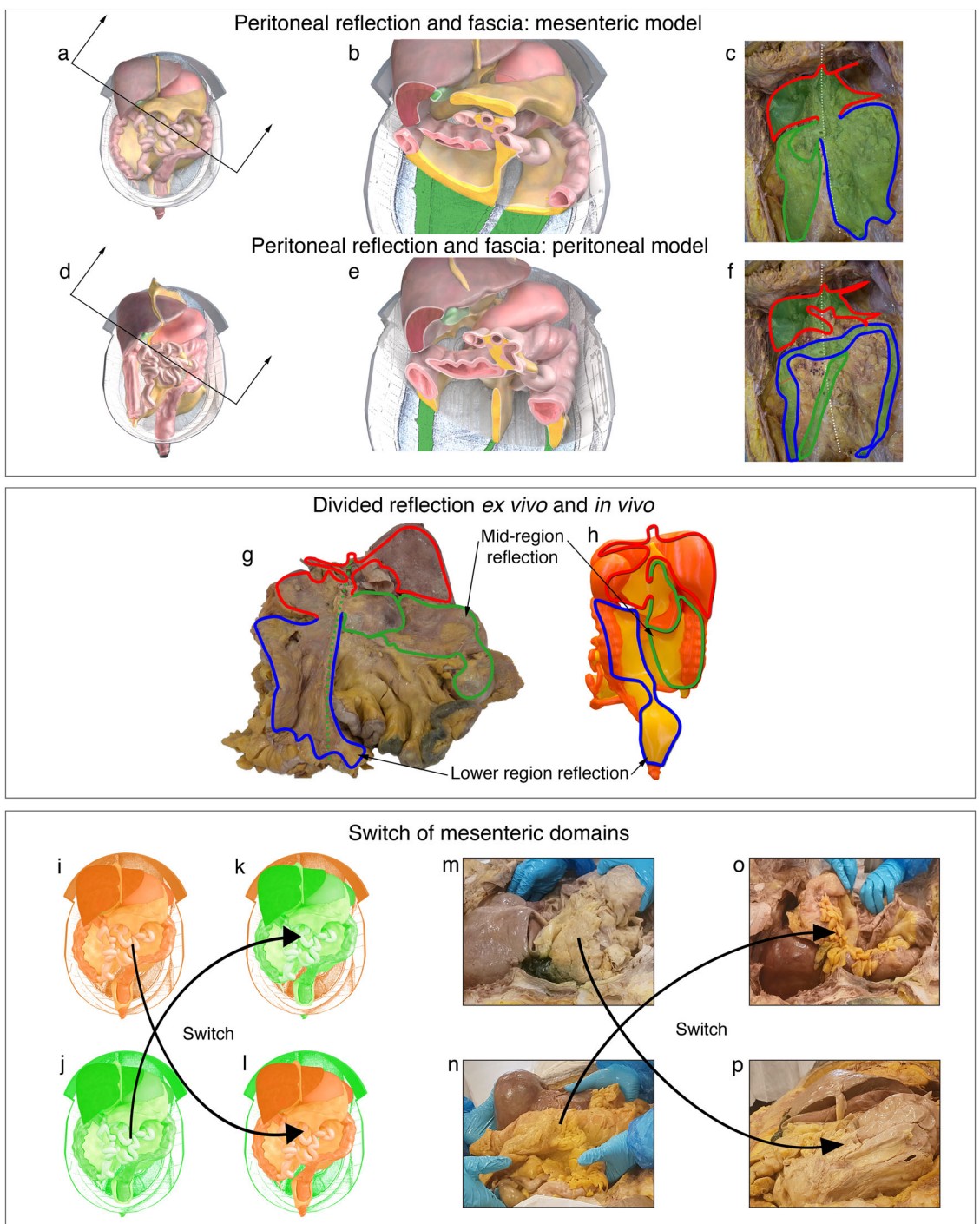

**Fig. 5 Mesenteric based anatomy.** Illustrations of the mesenteric model of the abdomen before (**a**), after (**b**) sectioning. **c** Photograph of the cadaveric non-mesenteric domain (schematic indicates the distribution of the reflection). Illustrations of the peritoneal model before (**d**), and after (**e**) sectioning. **f** Schematic indicating peritoneal based description of the mesentery/peritoneum. Photographs showing the posterior aspect of the ex vivo mesentery (**g**) and 3D printed mesentery (**h**) and reflection schematically superimposed. colour code: upper region (red), mid-region (green), lower region (blue). **i–p** Panel demonstrating switching of the mesenteric domain between cadavers in schematic (**i–l**) and cadaveric (**m–p**) formats.

connected to. The findings show that abdomino-pelvic organs are not primarily organised into peritoneal based domains. These are positioned in either the mesenteric or non-mesenteric domain. This organisation is maintained by a limited set of anatomical mechanisms. The study shows that all abdominopelvic contents are positionally layered on an anatomical foundation formed by visceral (i.e., mesenteric) and somatic (abdominal wall)

components. The organisation of the mesenteric and musculoskeletal frame corresponds to a fundamental order.

The mesenteric model is an alternative to the peritoneal model of abdominal anatomy. It resolves several questions in abdominal anatomy, e.g., the organisation of the peritoneum[37]. The peritoneal landscape is normally described in terms of sacs, fossae, recesses, gutters, pouches and cavities[30,31,38,39]. Peritoneal

borders are termed ligaments, folds, membranes or reflections. According to the mesenteric model, peritoneal spaces are the surface contours of both domains and the junction between these. Peritoneal borders are regions of reflection.

This organ-level study provides a framework for systematic study of the abdominal mesentery and conjugate organs[40]. Emerging evidence points to a similar organisation in the thorax[41,42]. If this suggestion held, combining findings above and below the diaphragm could have broad implications for our understanding of human form in general.

The findings provide diagnostic and therapeutic opportunities for a broad array of clinicians. They provide a developmental and anatomical foundation on which to reappraise human disease in general. They prompt reappraisal of abdominal anatomy and development in health and disease states, across the animal kingdom.

In summary, the mesentery is the organ in which all abdominal digestive organs develop and remain connected to. Abdomino-pelvic contents are organised into mesenteric and non-mesenteric domains. The mesentery and abdominal wall are the anatomical foundation of the abdomen and the organisation of these is a fundamental order on which abdominal anatomy is based. The foundation and its order provide a novel starting point for systematic characterisation of the abdomen in general.

## Methods

**Ethical approval**. Approval for this study was obtained from the local research ethics committee (University Hospital Limerick, Limerick, Ireland). Ethical approval for data received from international collaborators was obtained from their respective local research ethics committees. National legislation (Anatomy Act, Ireland 1832; Anatomy Act, Ireland 1871; Health Order, Ireland 1949; and Medical practitioners act, Ireland, 2007) governed the use of cadavers bequeathed to the National University of Ireland, Galway.

**Hardware/software specifications**. Hardware used included Apple (Apple Inc., Cupertino, California, United States) Mac Pro 5-1 workstation with the following specifications: Mac OS X *Lion* (v10.7.5) operating system, 2 ×6-Core Intel Xeon 2.4 GHz processors, 20GB 1333 MHz DDR3 RAM and an ATI Radeon HD 5770 1024 MB graphics card. A graphics tablet and pen (UG-1910B Graphic Tablet Monitor; UGEE, Guangzhou, China) were used for tracing. The suite of computing softwares used is summarised in Supplementary Table 1.

**Workflow involved in generating digital reconstructions of the mesentery and abdominal digestive organs**. Generation of digital models of the mesentery and digestive organs involved (a) data set acquisition, (b) alignment of sections, (c) pre-processing and segmentation of target organs, (d) 3D digital model development, (e) model stylization and (f) video of development (Supplementary Fig. 1).

(a) Dataset acquisition
Details related to the data sets used are summarised in Supplementary Tables 2 and 3. Datasets from the Carnegie Collection were utilised (https://www.ehd.org/virtual-human-embryo/)[43]. To generate a reconstruction of the adult human mesentery we used a data set from the Chinese Visible Human Datasets. The latter comprised a set of axial sections of an adult human, male cadaver[44–46].

(b) Alignment of sections
Individual slides were imported from each data-set (see above) into ImageJ2 and ordered craniocaudally as a stacked sequence. The voxel height (i.e., space between sections) and total stack height were adjusted to match the actual length of the segment of the specimen from which they were taken. When the histological data sets were first generated, each section was manually mounted onto individual slides. This means the alignment of each section relative to the section, differed. "Alignment" refers to the correction of this artefact and was conducted using the "register virtual stack" plugin in ImageJ2[47]. In the latter, the "rigid feature extraction" setting used. In that setting, the "rigid registration" and "translate and rotate" settings were adopted as previously described[48]. Post-alignment quality checks were performed. Where necessary, manual alignment was conducted using the transform "affine" function in the TrakEM2 plugin. To facilitate this, we cross-referenced the post-alignment stack with a photographic overview of the original specimen from which histological sections were obtained. In this manner, alignment artefacts related to curvature of the embryo were corrected for.

(c) Pre-processing and segmentation of target organs
Images were pre-processed to aid in the differentiation of individual structures. Due to limitations in computer random access memory (RAM) size, images were first resized to 300 dots per inch resolution and cropped to fit a maximum of 1 megabyte per cross-sectional image. Images were divided into red-green-blue channels at 32 bits per channel. Pixel size, dots per inch, quality and channel properties were standardised across datasets. Brightness, contrast and sharpness modifications were made using Adobe Lightroom. To further enhance contrast, 'adaptive histogram equalisation' was employed in the TrakEM2 plugin in ImageJ2[49–52].
Following pre-processing, the outline of structures was traced using a graphics tablet and pen (UG-1910B Graphic Tablet Monitor; UGEE, Guangzhou, China) as described by De Bakker et al. and Hikspoors et al.[53–55]. The threshold paint brush setting was used as described by Saalfeld et al.[47]. In most settings a clear demarcation was apparent between adjoining structures (e.g., mesentery and peritoneal cavity). The threshold based brush setting accurately reproduced the boundary between these structures, in visual format. In a limited number of settings demarcation was not immediately clear-cut. In these cases, boundaries were localised by consensus between investigators then manually depicted using a standard paint brush mode.

(d) 3D digital model development
3D reconstructions were generated in polygon mesh form, using the TrakEM2 plug-in (in ImageJ2). The "set scale" function was used in conjunction with original scale data to ensure accurate scaling. An ".obj" format polygon mesh was generated for each organ.

(e) Model stylizing
Meshes were imported into Cinema4D and aligned to a reference anatomical landmark (origin of the superior mesenteric artery (SMA)). A composite model involving all organs was developed. Following this, individual meshes (i.e., organs) were exported to ZBrush for stylizing. Clear-cut artefacts (e.g., disruption of tissues during preparation) were removed and individual regions were highlighted and coloured in a knowledge-driven, uniform manner (i.e., stylized). After modelling, all models were re-imported from ZBrush back to Cinema4D.

(f) Video of mesenteric and digestive organ development
A video depiction of mesenteric (and digestive organ) development was developed as follows (Supplementary Atlas Sections 4, 5). A model of the full term foetal mesentery was morphed back to the preceding stage using the "shrink-wrap" deformer in C4D. This ensured that both models contained an equal number of polygons. The approach was then repeated for all stages back to CS 13. This generated a series of models each with the same number of polygons. The "Pose-morph" tool was then used to generate an animation of development from CS 13 to the full term stage. A timeline was introduced using "key-frames." A camera was then introduced to generate a visual depiction of the animation over the time-line imposed. This generated a video depiction of the development of the mesentery from the CS 13 to full term stage. The overall process was repeated for all abdominal digestive organs. In this manner a video was generated depicting the development of all abdominal digestive organs, and their spatial relations, during the developmental program. To further illustrate the spatial relations between organs a boolean subtraction was introduced in C4D at select time points. This enabled us progressively subtract mesentery to expose underlying developing organs.

To simplify the appearance of the development of the mesentery additional models were generated in which small bowel folding was reduced. These models were processed as described above to generate an additional explanatory video depiction of mesenteric development (Supplementary Atlas Section 4). Multiple cameras were employed to provide several different viewpoints.

**3D printed models and silicone moulds of digestive organs**. 3D files (.obj,.stl) were prepared using Cinema4D R20 and Zbrush. Two 3D printers were used to print small-scale (Ultimaker 2, Ultimaker, Geldermalsen, Netherlands) and large-scale (Raise3d, Pro2 Plus, Irvine, CA, USA) models. For large-scale models, Polylactic acid (sized 1.75 mm) was used to generate 3D models. The printed layer height was 0.2 mm with an infill density of 15%. For small-scale models, polylactic acid (2.85 mm) was used with a printing height of 0.2 mm and an infill density of 10%. To generate silicone models, a large scale polyglactic acid mould was first 3D printed. Moulds were maintained in apposition using a wooden frame secured with metal brackets. Silicone was then poured through an aperture in the mould. After setting, the moulds were disassembled and the silicone model removed.

**Dissection of cadavers**.

(a) Adult human cadavers:
Adult human cadavers ($n = 6$) were used for dissection. Cadavers with documented abdominal disease or structural abnormalities, were excluded. Cadavers had been embalmed in formalin-based solution (12 L water, 6 L glycerin, 6 L methanol, 2.4 L 37-41% formalin solution and 2 L phenol). All dissection was recorded using a Canon 550D camera (Canon Inc., Tokyo, Japan). The dissection technique is demonstrated in Supplementary Atlas Section 6.

(b)  Animal cadavers:

Frozen, animal cadavers (chimpanzee, pygmy marmoset, raccoon, dog, cat, goat, sheep, bullock, deer, pheasant, snip (common and jack species), woodcock, teal duck, mallard duck were dissected using a "mesenteric-based" approach similar to that demonstrated in Supplementary Atlas Sections 6.0–6.15. Accordingly, the mesentery, and connected organs were excised *en bloc* and inspected. Photographic records were generated as described above (Supplementary Atlas Section 7).

**Scanning electron and light microscopy**. Samples of the peritoneal reflection were harvested from each of the following regions: ileocaecal reflection, right lateral reflection, left lateral reflection, sigmoid reflection. A 1-cm$^2$ area of peritoneal reflection was excised sharply from each. All samples were then primarily fixed in a solution containing: 2% glutaraldehyde, 2% paraformaldehyde and 0.1 M of sodium cacodylate/HCl buffer. Specimens were then all dehydrated through a graded series of ethanol solutions of increasing concentrations and then mounted on metal studs. After drying, a gold splutter coat was applied. SEM analysis was performed using a Hitachi S2600N Variable Pressure Scanning Electron Microscope (Hitachi, Tokyo, Japan).

Light microscopy was performed using a Leica DM750 light microscope with an ICC50HD camera attachment. Tissue from each region was formalin fixed and paraffin wax embedded. Four μm sections were deparaffinized, hydrated then stained with haematoxylin and eosin.

**Characterisation of brightness values**. To quantitate and display brightness values in mesenteric and non-mesenteric tissues, data set photomicrographs were converted to grayscale. Histograms depicting brightness values (see Supplementary Data 1) in delineated regions were generated using the histogram function in ImageJ2. To depict differences in brightness values between tissues, overlapping histograms were generated.

**Reconstruction of in vivo mesenteric vasculature using abdominal cross sectional imaging**. Ethical approval was obtained from the Ethics Committee at the University of Limerick Hospital Group. Having acquired informed consent, computerised axial tomographic abdominal data sets were obtained on ten patients who had previously undergone abdominal surgery, and were known to have normal intestino-mesenteric anatomy. These were imported into Osirix and branches of the superior mesenteric artery (SMA) were reconstructed. The inferior pancreaticoduodenal artery was taken as a marker of the mesoduodenal region of the mid-region fold. The first jejunal branch of the SMA marked the proximal mesojejunum. The middle colic artery marked the junction between the central and peripheral zones of the mid region fold. Collectively, the inferior pancreatico-duodenal and first jejunal arteries (on the right) and the middle colic (on the left) marked the junction between the central and peripheral zones of the mid-region fold. The position of each vessel relative to the SMA was then determined.

**Conal model of mesenteric morphology**. A cone-based geometric representation of the mesentery was developed in which each cone was based on a major arterial vessel. The apex of the cone corresponded to the origin of the artery. Cones were based on the coeliac trunk, SMA, middle and left colic and the inferior mesenteric artery. Conal modelling predicts that the trajectory of the periphery of the mesentery will approximate to an ellipse centrally, but will deviate from this at junctions with successive cones. To test this, we used the intestine as a marker of the periphery of the mesentery. For each point along the intestine, we established (1) its distance from a fixed point and (2) distance along the arterial axis supporting that point. We applied this to the intestine of the mid-region fold (i.e. from mid-duodenal to splenic flexure level). This gave x and y values for points along the intestine. We depicted the course of the intestine (i.e. the periphery of the mesentery) as a line in the x,y plane. The line represented the actual course of the periphery of the mesentery. From this line, we generated a near-fit ellipse and calculated "a" and "b" parameters for that ellipse. Using these parameters, and *actual* x values we calculated a *predicted* y value for each point (see Supplementary Data 2). Combining actual x and predicted y values we generated a *predicted* line. This represented the predicted course of the periphery of the mesentery, if it were to follow an ellipse whose "a" and "b" parameters were established. We then directly compared the actual and predicted lines and in so doing compared the actual course of the periphery of the mesentery with a predicted course based on a near-fit ellipse.

**Atlas of the mesentery**. An online Atlas of the Mesentery (http://mesentery.ie) was compiled. In each section, interactive 3D models were included to enable the reader directly explore the shape of the mesentery and related structures. Section 1 includes 3D digital reconstructions of the mesentery at each stage of development. Section 2 includes 3D reconstructions relevant to all figures. Section 3 includes a video depiction of the development of the mesentery. At select times the video is paused, and the model progressively sectioned through to demonstrate contained structures. Section 4 includes a video depiction of the development of the main body of the mesentery, and in which coil/buckling of the small intestinal region of

the mesentery has been excluded. Section 5 includes videos demonstrating the development of individual abdominal digestive organ, in adjoining mesentery. Section 6 includes a series of narrated videos demonstrating step-wise dissection of the human ex vivo mesentery and mesenteric domain. This section provides a manual of how to excise the intact mesentery (and hence mesenteric domain of the abdomen). Section 7 is a photographic record of animal cadavers during dissection.

**Statistics and reproducibility**. Statistical comparisons were not conducted. The numbers and nature of the specimens used are detailed in the Supplementary tables. Supplementary Note 1 addresses the question of the validity of the models generated.

**Reporting summary**. Further information on research design is available in the Nature Research Reporting Summary linked to this article.

## Data availability
Data from reconstructions and dissections that support the findings of the study are available in the Atlas of the mesentery (http://mesentery.ie). The data that support the findings of the study are available from the corresponding author upon request.

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

## Acknowledgements

We thank Marco C. DeRuiter (Leiden University Medical Centre, Leiden) for supplying digitized histological sections for analysis.

## Author contributions

K.G.B.—data collection, analysis, 3D reconstruction and manuscript writing. D.W.—3D reconstruction, modelling and cadaveric dissection. L.G.W.—animal dissections and manuscript review. D.M.C.—3D printing and modelling. M.F.U.—cadaveric dissection. R.M.—scanning and analysis of histological sections. J.H.—scanning, digitization of histological sections. W.L.—analysis of histological sections and review of manuscript and manuscript revision and manuscript revision. Y.W.—performed scanning and analysis of cross-sectional human cadaveric dataset and manuscript revision. X.Q.Z.—performed scanning and analysis of cross-sectional human cadaveric dataset and manuscript revision. S.X.Z.—performed scanning and analysis of cross-sectional human cadaveric dataset and manuscript revision. P.B.—performed comparative dissection study across species. C.P.D.—manuscript editing. I.S.O.—performed cadaveric dissection. C.B.P.—revision of manuscript. M.J.S.—analysis and reconstruction of human CT scans. T.G.S.—analysis and reconstruction of human CT scans. M.E.L.—comparative anatomy dissections. H.D.B. performed comparative anatomy dissections. P.D.—cadaveric dissection and writing of manuscript. K.W.M.—data interpretation and writing of manuscript. J.C.C.—project design and supervision, data interpretation and manuscript writing. All authors reviewed the manuscript and approved the final version of the manuscript.

## Competing interests

The authors declare no competing interests.
