## [Transparent Peer Review File · Communications Biology]

Reviewers' comments:

Reviewer #1 (Remarks to the Author):

In the last decade, the work of Professor Coffey and his team has significantly influenced our understanding of mesenteric and peritoneal anatomy. Their work in seeking to reconcile divergent anatomical and surgical views has significant implications for a wide range of readers, including anatomists (human and comparative anatomy), embryologists, cell and molecular biologists and surgeons: note that the list is by no means exhaustive.

Here, the authors have reconstructed and interpreted the developing mesentery from digitized datasets. They have used several approaches that are presented in extenso as Supplementary Information (including videos of key events in mesenteric development and a detailed step-by-step cadaveric dissection) and Model Validation to characterise the relationship between the mesentery and the posterior abdominal wall. They conclude that there is one mesentery within the abdomen within which all abdominal digestive organs develop and to which the latter remain connected; that all abdominopelvic organs are organised into two, discrete anatomical domains, mesenteric and non-mesenteric; that the abdomen comprises a visceral (mesenteric) and somatic (musculoskeletal) frame; that mesenteric continuity and direct organ connectivity is retained in the adult. Having dissected various animal cadavers, they report that a similar organisation occurred across a range of animal species.

Comments

1. Given the extensive body of work that Professor Coffey and his team have already published in this area, it would be helpful to have an indication of the novel findings described here.
2. An agreed terminology is essential in any field, although sometimes difficult to achieve (particularly in anatomy): consensus clarifies, whereas disagreement confuses. For the benefit of a wider readership, some of whom may be less familiar with the previous work of Professor Coffey and his colleagues, I suggest that it would be helpful if the authors would define the terms 'mesentery' and 'peritoneum' and establish their relationship at first meeting. For example, in lines 59-61 we are told that '...a network of peritoneal derivatives... include mesenteries, duplicatures of peritoneum...' Later, 'peritoneal-like reflection' (line 160) and 'peritonealised surfaces' would benefit from explanation. The caption for Fig 5f refers to '... a peritoneal based description of the mesentery/peritoneum'. I note that in the Terminologia Anatomica version 2 (2019), the peritoneum is referred to as the 'peritoneum of the mesentery'.
3. Lines 63 and 64: 'Although the peritoneal model is the standard model of abdominal organisation, it is unclear how IT develops'. To what does my capitalised 'IT' refer?
4. Lines 65 et seq: Given that the peritoneal model does not explain current surgical techniques or radiological depictions of the abdomen, it would be helpful to add text here summarising the reason(s) why the peritoneal model has proved to be inadequate (e.g. mesenteric-based strategies for managing intestinal cancer offer better outcomes).
5. Line 94: Define CS as 'Carnegie stage' in text – I appreciate this is defined in the caption for Fig 1a, but suggest that it should also be defined in parentheses in the text at first meeting.
6. Lines 136, 219: renaissance should have an initial upper case 'R' in both cases.
7. Line 151: define mesoduodenum and mesojejunum (note that neither term appears in Terminologia Anatomica 2, 2019).
8. Line 220: coelomata should have an upper case initial 'C'.
9. Fig 3a: I recommend adding a colour code in the caption, especially for the pancreas.
10. In Figs 3f, g: the label 'Mesorectum' lacks leader lines.
11. In the online Atlas of the Mesentery (https://urldefense.proofpoint.com/v2/url?u=http-3A__mesentery.ie&d=DwIF-g&c=vh6FgFnduejNhPPD0fl_yRaSfZy8CWbWnIf4XJhSqx8&r=OF9Zm-rVFV8VdjQ298Jy6FG8xtoQTCTFebwS-ZvWixc&m=_6DCFnxIgdKnnQhrXehqxHRv85WIKUWVrBoVBeq5-U&s=d3H3a6dOHjVFWHxDloMF58UY5FG6fspiaHrxDG44o2I&e=), under 'Manual of mesenteric dissection' there is a page (09.40-10.01) that reads 'To be replaced with video' – presumably this text should be deleted/replaced?

Reviewer #2 (Remarks to the Author):

The manuscript used human embryos (from CS 13) and fetuses to adult human cadavers and different animal cadavers to describe mesentery development using serial sections and wonderful 3D technique. They want to prove all abdominopelvic organs organized into mesenteric and non-mesenteric domain. The paper was well designed, and based on explicit anatomical images that reader can easy understand.

In my opinion, the submitted manuscript can be accepted for publication in Communications Biology without modification.

Additional comment:

1) We know mesentery anatomy is very important for clinical work, like as endoscope operation.
2) There were few papers about mesentery structure of 3D reconstruction using human embryo and fetus.

Soffers JH, Hikspoors JP, et al. The growth of the human intestine and its mesentry. *BMC Dev Biol.* 2015; 15:31. Doi:10.1186/s12861-01509981-x.

Nakamura T, Yamada S. et al. Three-dimensional morphogenesis of the omental bursa from four recesses in stage human embryos. *J Anat.* 202; 237(1):166-175. Doi: 10.1111/joa.13174.

3) They used six step in general digital reconstructions of mesentery and abdominal organs. a) data set acquisition; b) alignment of sections; c) pre-processing and segmentation of target organs; d) 3D digital model development; e) model stylization; f) video of development.

It is important in step b), c) and e) to get correct and beautiful 3D reconstruction results.

In here, they also used data Chinese Visible Human Datasets (leader: Prof. Shao-Xiang Zhang). Prof. Zhang is chairman of CSAS and ISDM. He lead Chinese Visible Human project for long time. They have very powerful technicians with good skill in the world, and they have great computer skills in sectional image alignment and 3D reconstruction.

According to their reconstruction results, it is helpful for surgeons to correctly understand the structure and application of human mesentery.

Reviewer #3 (Remarks to the Author):

Dear Authors,

Thank you for this submission which adds a great visual representation of the mesentery, and its anatomical relations and progress during development.

It is a very well explained and detailed description using a standard resource of the Carnegie Stages.

I had been invited to comment on the following areas specifically:

Image Analysis:

This represents commonly used digital software tools to undertake visualisation and reconstruction. It clearly demonstrates

3D Digital Anatomical Constructions:

These are beautifully digitised and represent a clear explanation for the development and structure of the mesentery and related abdominal structures.

Geometric Anatomical Representation:

This represents and excellent visualisation methodology to explain the structures, relations, development, positions and changes that occur during embryological development.

One thing which would have been good to see would be an animation of this to allow an all angle viewpoint for the reader. This type of tool would allow a better 3D reconstruction and visualisation

for those wanting to learn more about this, both for students, teaching and clearer views. Whilst this is great to see with the colouring, segmentation and digital reconstruction, this is an area where learners to this field struggle with the most. A section explaining this, and the limitations of this approach would add the final details to what is a well written manuscript.

The discussion section is very brief and could detail the wide and varied opportunities for further research and applications.

Thank you for the opportunity to review this.

Re: COMMSBIO-21-0556A

Date: 17_6_21

Dear Reviewers,

Thank you for reviewing this manuscript.

We have addressed all comments by each reviewer in the following.

We note that reviewer two recommended the manuscript be published without modification. It would seem from that, that the comments the reviewer made do not require addressing.

Many thanks again for reviewing the manuscript.

Best wishes

Professor J Calvin Coffey
MB., B.MedSci., B.Sc., PhD., FRCSI,
Head of Department of Surgery
University of Limerick Hospital Group.
Professor and Chair of Surgery,
School of Medicine,
University of Limerick.
Consultant General and Colorectal Surgeon,
UHLG, Limerick, Ireland.

Reviewer #1

In the last decade, the work of Professor Coffey and his team has significantly influenced our understanding of mesenteric and peritoneal anatomy. Their work in seeking to reconcile divergent anatomical and surgical views has significant implications for a wide range of readers, including anatomists (human and comparative anatomy), embryologists, cell and molecular biologists and surgeons: note that the list is by no means exhaustive.

Here, the authors have reconstructed and interpreted the developing mesentery from digitized datasets. They have used several approaches that are presented in extenso as Supplementary Information (including videos of key events in mesenteric development and a detailed step-by-step cadaveric dissection) and Model Validation to characterise the relationship between the mesentery and the posterior abdominal wall. They conclude that there is one mesentery within the abdomen within which all abdominal digestive organs develop and to which the latter remain connected; that all abdominopelvic organs are organised into two, discrete anatomical domains, mesenteric and non-mesenteric; that the abdomen comprises a visceral (mesenteric) and somatic (musculoskeletal) frame; that mesenteric continuity and direct organ connectivity is retained in the adult. Having dissected various animal cadavers, they report that a similar organisation occurred across a range of animal species.

Response: We would like to thank the reviewer for the above points.

Comments:

Comment: 1. Given the extensive body of work that Professor Coffey and his team have already published in this area, it would be helpful to have an indication of the novel findings described here.

Response: Our previous work focussed on the mesentery distal to the duodenojejunal flexure. That work, in conjunction with illustrations from Renaissance anatomists and artists, pointed to (but did not prove) that the abdomen is organised along mesenteric and not peritoneal lines. The present body of work addressed the entire mesentery and the developmental program by which the mesentery acquires the shape apparent in the adult setting. In characterising these, it uncovered direct proof that the abdomen in the adult human, comprises two distinct compartments (the mesenteric and non-mesenteric domain) and that it (the abdomen) is not organised along peritoneal lines.

Our previous work clarified the positional anatomy of the small and large intestine; the anatomical position of these corresponds to position relative to the adjoining mesentery. The work in the present submission clarified the positional anatomy of **all** abdominal digestive organs. It also clarified the positional anatomy of the arterial inflow and venous drainage of all organs of the abdominal digestive system. The work thus appears to resolve several challenges in abdominal anatomy. For example, the anatomical position of the pancreas has conventionally been described in terms of multiple, nearby organs. According to the findings of this study, the position of the pancreas can be described with reference to one organ, the

The findings also appear to resolve an age-old question in that they help clarify the nature of the peritoneum as well as that of the mesentery. The two terms have often been used interchangeably as reflected in the conventional definition of the mesentery (i.e. it was conventionally described as a double fold of peritoneum). Terminologica Anatomica 2 lists the mesentery under the heading of “Peritoneal structures.” The presents findings confirm that the parietal peritoneum is the surface lining of the free (i.e. unopposed) surface of the abdomen. They demonstrate that the visceral peritoneum corresponds to the free surface of the mesenteric domain. They also show that the peritoneal refection is the junction between both parietal and visceral peritoneum.

This in turn explains several aspects of abdominal anatomy that continue to challenge scientists and clinicians. The sacs, folds, recesses, fossae, cavities and pouches of the peritoneal cavity are explained as being regions of the surface of the mesenteric and non-mesenteric domain and the junction between both. The reflections, ligaments, membranes and folds of the peritoneal landscape are regions of the reflection of the peritoneum between the parietal peritoneum (overlying the non-mesenteric domain) and the visceral peritoneum (overlying the mesenteric domain).

The work includes a comparative anatomical section, which points to similar findings in species in the animal kingdom. It also includes a modelling section to address potential mechanisms by which the adult shape of the mesentery, and adjoining intestine, arise.

Changes to manuscript:

Abstract:

Here we **confirm** that, within the abdomen, there is one mesentery in which all abdominal digestive organs develop and remain connected to.

Comment 2. An agreed terminology is essential in any field, although sometimes difficult to achieve (particularly in anatomy): consensus clarifies, whereas disagreement confuses. For the benefit of a wider readership, some of whom may be less familiar with the previous work of Professor Coffey and his colleagues, I suggest that it would be helpful if the authors would define the terms ‘mesentery’ and ‘peritoneum’ and establish their relationship **at first meeting.** For example, in lines 59-61 we are told that ‘.a network of peritoneal derivatives... include mesenteries, duplicatures of peritoneum...’

Response: We have commenced the introduction with the following sentences, in keeping with this suggestion.

Until recently, the mesenteries were considered peritoneal structures and defined as a double fold of peritoneum connecting some regions of intestine to the posterior abdominal wall. The peritoneum is defined as the serous membrane lining of the abdominal cavity and contained organs.

Comment 3: Later, ‘peritoneal-like reflection’ (line 160) and ‘peritonealised surfaces’ would benefit from explanation.

Response: These have been addressed as follows:

“.....A fascia occurred between apposed surfaces and a reflection that resembled peritoneum, marked the peripheral limit of apposition (Fig. 5)....”

“.....The reflection was located at the periphery of the mesenteric domain where it bridged the peritonealised (i.e. mesothelial) surfaces of both domains....”

Comment 4: The caption for Fig 5f refers to ...’ a peritoneal based description of the mesentery/peritoneum’.

Response: We hope that it is okay to leave the caption for Fig 5c unaltered as the figure presents a schematic of the peritoneal-based model of abdominal anatomy. The aim here was to enable the reader directly compare the mesenteric (Figs 5a-c) and peritoneal (Figs 5d-f) models of abdominal anatomy.

Comment 5: I note that in the Terminologia Anatomica version 2 (2019), the peritoneum is referred to as the ‘peritoneum of the mesentery’.

Response: TA2 lists the mesentery under the heading of “Peritoneal structures.” We mention the convention at the start of the introduction, and in the first references of the bibliography. Given this, we have omitted referencing TA2 directly. We would be happy to change this if it is recommended we do so.

Comment 6: Lines 63 and 64: ‘Although the peritoneal model is the standard model of abdominal organisation, it is unclear how IT develops’. To what does my capitalised ‘IT’ refer?

Response: This has been clarified as follows:

Introduction:

“....Although the peritoneal model is the standard model of abdominal organisation, it is unclear how this model develops. The model does not explain current surgical techniques or radiological depictions of the abdomen¹¹⁻¹⁴....”

Comment 7: Lines 65 et seq: Given that the peritoneal model does not explain current surgical techniques or radiological depictions of the abdomen, it would be helpful to add text here summarising the reason(s) why the peritoneal model has proved to be inadequate (e.g. mesenteric-based strategies for managing intestinal cancer offer better outcomes).

Response: We have included a line to this effect.

Introduction:

“....For example, the surgical techniques of total mesorectal excision and complete mesocolic excision are mesenteric-based, and are associated with improved outcomes for patients undergoing surgery for rectal or colon cancer respectively^{12,13}....”

Comment 8. Line 94: Define CS as ‘Carnegie stage’ in text – I appreciate this is defined in the caption for Fig 1a, but suggest that it should also be defined in parentheses in the text at first meeting.

Response: this has been corrected

Comment 9. Lines 136, 219: renaissance should have an initial upper case ‘R’ in both cases.

Response: These have been corrected.

Comment 10. Line 151: define mesoduodenum and mesojejunum (note that neither term appears in Terminologia Anatomica 2, 2019).

Response: These have been defined in the text. We would argue in favour of these terms being included in future iterations of TA.

Comment 11. Line 220: coelomata should have an upper case initial ‘C’.

Response: This has been corrected.

Comment 12. Fig 3a: I recommend adding a colour code in the caption, especially for the pancreas.

Response: This has been corrected and a color code has been included

Comment 13. In Figs 3f, g: the label ‘Mesorectum’ lacks leader lines.

Response: These have been included

Comment 14. In the online Atlas of the Mesentery (https://urldefense.proofpoint.com/v2/url?u=http-3A_mesentery.ie&d=DwIF-g&c=vh6FgFnduejNhPPD0fl_yRaSfZy8CWbWnIf4XJhSqx8&r=OF9Zm-rVFV8VdjQ298Jy6FG8xtoQTCtFebwS-ZvWixc&m=_6DCFnXlgdKnnQhrXehqxHRv85WIIKUWVrBoVBeq5-U&s=d3H3a6dOHjVFWHxDloMF58UY5FG6fspiaHrxDG44o2I&e=), under ‘Manual of mesenteric dissection’ there is a page (09.40-10.01) that reads ‘To be replaced with video’ – presumably this text should be deleted/replaced?

Response: We thank the reviewer for bringing this error to our attention, we have included the video initially intended for that section.

Reviewer #2

Comment: The manuscript used human embryos (from CS 13) and fetuses to adult human cadavers and different animal cadavers to describe mesentery development using serial sections and wonderful 3D technique. They want to prove all abdominopelvic organs organized into mesenteric and non-mesenteric domain. The paper was well designed, and based on explicit anatomical images that reader can easy understand. In my opinion, the submitted manuscript can be accepted for publication in Communications Biology without modification.

Response: We thank the reviewer from these comments and the recommendation the manuscript be accepted without modification.

Additional comment:

1) We know mesentery anatomy is very important for clinical work, like as endoscope operation.

2) There were few papers about mesentery structure of 3D reconstruction using human embryo and fetus. Soffers JH, Hikspoors JP, et al. The growth of the human intestine and its mesentery. BMC Dev Biol. 2015; 15:31. Doi:10.1186/s12861-01509981-x.

Nakamura T, Yamada S. et al. Three-dimensional morphogenesis of the omental bursa from four recesses in stage human embryos. J Anat. 202; 237(1):166-175. Doi: 10.1111/joa.13174.

3) They used six step in general digital reconstructions of mesentery and abdominal organs. a) data set acquisition; b) alignment of sections; c) pre-processing and segmentation of target organs; d) 3D digital model development; e) model stylization; f) video of development.

It is important in step b), c) and e) to get correct and beautiful 3D reconstruction results.

In here, they also used data Chinese Visible Human Datasets (leader: Prof. Shao-Xiang Zhang). Prof. Zhang is chairman of CSAS and ISDM. He lead Chinese Visible Human project for long time. They have very powerful technicians with good skill in the world, and they have great computer skills in sectional image alignment and 3D reconstruction.

According to their reconstruction results, it is helpful for surgeons to correctly understand the structure and application of human mesentery.

Reviewer #3

Dear Authors,

Thank you for this submission which adds a great visual representation of the mesentery, and its anatomical relations and progress during development. It is a very well explained and detailed description using a standard resource of the Carnegie Stages. I had been invited to comment on the following areas specifically:

Image Analysis:

Comment 1 This represents commonly used digital software tools to undertake visualisation and reconstruction. It clearly demonstrates 3D Digital Anatomical Constructions:

These are beautifully digitised and represent a clear explanation for the development and structure of the mesentery and related abdominal structures.

Response: We thank the reviewer for these very generous comments.

Comment 2: Geometric Anatomical Representation: This represents an excellent visualisation methodology to explain the structures, relations, development, positions and changes that occur during embryological development.

Response: We again thank the reviewer for these very positive comments.

Comment 3: One thing which would have been good to see would be an animation of this to allow an all angle viewpoint for the reader. This type of tool would allow a better 3D reconstruction and visualisation for those wanting to learn more about this, both for students, teaching and clearer views. Whilst this is great to see with the colouring, segmentation and digital reconstruction, this is an area where learners in this field struggle with the most. A section explaining this, and the limitations of this approach would add the final details to what is a well written manuscript.

Response: We thank the reviewer for this point. In the www.mesentery.ie website, we have supplied models that the reader can view. These are interactive models that the reader can then rotate, scale and view from several different vantage points. Unfortunately it is not possible for us to provide an all angle animation, but we hope the models included at www.mesentery.ie will provide the reader with the capacity to achieve an all angle view of static models.

Comment 4: The discussion section is very brief and could detail the wide and varied opportunities for further research and applications.

Response: We have now included some additional sentences in line with this recommendation. We did not include these in the first submission in view of the limitation on word counts.

Changes made:

The following paragraph has been added to the “Discussion” section:

“...The findings provide diagnostic and therapeutic opportunities for a broad array of clinicians. They provide a developmental and anatomical foundation on which to reappraise human disease in general. They prompt reappraisal of abdominal anatomy and development in health and disease states, across the animal kingdom...”

REVIEWERS' COMMENTS:

Reviewer #1 (Remarks to the Author):

The authors have responded to all of the suggestions/comments in my original review of their very nice paper. I note their response to my comment 5 and am happy that TA2 is not referenced directly.

I have two very minor points:

Line 72: the peritoneal-model: for consistency across the manuscript, the hyphen should be deleted in this new text.

Line 242: 'The findings clarify the nature of the mesentery; IT AS the organ in which...' Please clarify 'it as' (my caps).

I suggest EITHER:

The findings clarify the nature of the mesentery as the organ in which all abdominal digestive organs develop and remain connected to.

OR

The findings clarify the nature of the mesentery. It is the organ in which all abdominal digestive organs develop and remain connected to.

Reviewer #3 (Remarks to the Author):

I am happy with these comments and additions and thank the authors for such a swift and comprehensive review changes.

I recommend to publish with no further changes.

EDITOR NOTE:

I'd also like to inform you that while this manuscript was not sent back to the original Reviewer 2 to review this revision, after the initial revise decision was sent, this reviewer followed up with additional minor points to consider. I will include this text below:

New Comment

The authors reconstructed abdominal structures from CS 13 to 12 week human embryos and fetuses for study mesenteric development. They found that the mesentery is continuous and composite at all stages, and that all abdominal digestive organs developed in or on the mesentery. In addition, they observed the relationship between mesentery and abdominal wall with adult cadavers. Moreover, they compared with some of animal cadavers and confirmed the anatomical foundation of the human abdomen, which is applicable to a range of animal species. Based on this, they concluded that the mesentery is the organ with which all abdominal digestive organs develop and remain connected to.

The design of the manuscript is reasonable, and the image data is reliable and rich. By using powerful computer technology, the author has established 3D images and 3D models which are easy understand. To provide education and reference resource for the research of human development, anatomy, and clinical surgery.

In my opinion, the submitted manuscript can be accepted for publication in "Communications Biology" with minor revise.

Minor revise:

Attention to the format, there are many double space in the text.

Standardized anatomical terminology should be used. Like as "peritonealised", "oesophago-gastric"

etc.

Each references should be arranged as journal form.

Line 46: ...(i.e. musculoskeletal)...

Line 60: ...mesenteries, duplicates of ...

In figure 5a, it's better to have space between pictures.

Re: COMMSBIO-21-0556A

Date: 3_7_21

Dear Reviewers,

Thank you for reviewing this manuscript and for your very generous comments.

We have made all recommended edits as outlined in the following.

Many thanks again for reviewing the manuscript.

Best wishes

Calvin

Professor J Calvin Coffey
MB., B.MedSci., B.Sc., PhD., FRCSI,
Head of Department of Surgery
University of Limerick Hospital Group.
Professor and Chair of Surgery,
School of Medicine,
University of Limerick.
Consultant General and Colorectal Surgeon,
UHLG, Limerick, Ireland.

Reviewer #1

Comment 1: The authors have responded to all of the suggestions/comments in my original review of their very nice paper. I note their response to my comment 5 and am happy that TA2 is not referenced directly.

Response: We thank the reviewer for this.

Comment 2: I have two very minor points: Line 72: the peritoneal-model: for consistency across the manuscript, the hyphen should be deleted in this new text.

Response: We have made the recommended change across the text.

Comment 2: Line 242: 'The findings clarify the nature of the mesentery; IT AS the organ in which...' Please clarify 'it as' (my caps).

I suggest EITHER:

The findings clarify the nature of the mesentery as the organ in which all abdominal digestive organs develop and remain connected to.

OR

The findings clarify the nature of the mesentery. It is the organ in which all abdominal digestive organs develop and remain connected to.

Response: We have gone with the second suggestion in the revised manuscript.

Reviewer #2

Comment: "...The design of the manuscript is reasonable, and the image data is reliable and rich. By using powerful computer technology, the author has established 3D images and 3D models which are easy understand. To provide education and reference resource for the research of human development, anatomy, and clinical surgery..."

Response: We thank the reviewer for their very generous comments regarding the paper.

Comments: Attention to the format, there are many double space in the text.

Response: We have focussed closely on formatting all submissions in line with the requirements of the journal as detailed at style and formatting checklist.

Comments: Standardized anatomical terminology should be used. Like as "peritonealised",

"oesophago-gastric"

Response: We have adhered to the terminology used in *Terminologica Anatomica* version 2 (published 2019) throughout the manuscript.

Comments Each references should be arranged as journal form.

Response: We have adhered to the formatting requirements of the journal as detailed at style and formatting checklist.

Comment: Line 46: ...(i.e. musculoskeletal)...

Response: This change has been made.

Comment: Line 60: ...mesenteries, duplicates of ...

Response: We defined the mesenteries at the start of the introduction and in line with *Terminologica Anatomica* version 2. As a result, we have omitted inclusion of the word “duplicates” after the word “mesenteries” in line 60.

Comment: In figure 5a, it’s better to have space between pictures

Response: Figure 5a is a schematic representation of the relationship between the mesenteric and non-mesenteric domains. For the purposes of accuracy these domains are apposed. This is because an anatomical space is not apparent between the domains. To retain anatomical accuracy we have not altered the image.

Reviewer 3:

I am happy with these comments and additions and thank the authors for such a swift and comprehensive review changes. I recommend to publish with no further changes

Response: We thank the reviewer for this.